# Knee Arthrodesis for Periprosthetic Knee Infection: Fusion Rate, Complications, and Limb Salvage—A Systematic Review

**DOI:** 10.3390/healthcare12070804

**Published:** 2024-04-07

**Authors:** Michele Mercurio, Giorgio Gasparini, Erminia Cofano, Andrea Zappia, Filippo Familiari, Olimpio Galasso

**Affiliations:** 1Department of Orthopaedic and Trauma Surgery, Magna Graecia University, 88100 Catanzaro, Italy; michele.mercurio@unicz.it (M.M.); gasparini@unicz.it (G.G.); andreasisto.zappia@studenti.unicz.it (A.Z.); filippofamiliari@unicz.it (F.F.); galasso@unicz.it (O.G.); 2Research Center on Musculoskeletal Health, MusculoSkeletal Health@UMG, Magna Graecia University, 88100 Catanzaro, Italy; 3Clinica Ortopedica Department, San Giovanni di Dio e Ruggi D’Aragona University Hospital, 84100 Salerno, Italy; 4Department of Medicine, Surgery and Dentistry, University of Salerno, Baronissi, 84081 Salerno, Italy

**Keywords:** knee arthrodesis, periprosthetic knee infection, intramedullary nail, external fixator, reinfection, fusion rate, amputation

## Abstract

The aim of this systematic review was to investigate the outcomes of knee arthrodesis (KA) after periprosthetic joint infection (PJI) of the knee. Differences in clinical outcomes and complication rates among the intramedullary nailing (IMN), external fixation (EF), and compression plating (CP) procedures were compared. A total of 23 studies were included. Demographics, microbiological data, types of implants, surgical techniques with complications, reoperations, fusion, and amputation rates were reported. A total of 787 patients were evaluated, of whom 601 (76.4%), 166 (21%), and 19 (2.4%) underwent IMN, EF, and CP, respectively. The most common causative pathogen was *coagulase-negative Staphylococcus* (*CNS*). Fusion occurred in 71.9%, 78.8%, and 92.3% of the patients after IMN, EF, and CP, respectively, and no statistically significant difference was found. Reinfection rates were 14.6%, 15.1%, and 10.5% after IMN, EF, and CP, respectively, and no statistically significant difference was found. Conversion to amputation occurred in 4.3%, 5%, and 15.8% of patients after IMN, EF, and CP, respectively; there was a higher rate after CP than after EF. The IMN technique is the most common option used for managing PJI with KA. No differences in terms of fusion, reinfection, or conversion-to-amputation rates were reported between IMN and EF. CP is rarely used, and the high amputation rate represents an important limitation of this technique.

## 1. Introduction

Periprosthetic joint infection (PJI) is a challenging complication after total knee arthroplasty (TKA), with an incidence ranging from 0.4% to 2% in primary TKA [1,2] and a recurrence rate after revision surgery ranging from 14% to 28% [3]. Different treatment approaches to eradicate PJI have been used, and these include long-term suppressive antibiotics, debridement and implant retention, polyethylene liner exchange, replacement of the implant in one or two stages, and salvage procedures such as knee arthrodesis (KA), resection arthroplasty, and amputation [4]. While each of these methods may be appropriate for a given patient, two-stage reimplantation has been reported to be the most successful method for treating TKA infections, with reported success rates of 88–96% [5]. KA represents a valid alternative to amputation in patients with multiple recurrent PJIs or when the PJI is accompanied by severe bone loss, an unreconstructible extensor mechanism, and poor soft-tissue coverage. However, high complication, non-union, and infection rates have been reported; in addition, the loss of the range of motion of the knee may be deterrent for patients [4]. It should also be considered that KA may provide functional results superior to those of resection arthroplasty and amputation, restoring a stable extremity and weight-bearing ability, while a prior contralateral KA, amputation, or ipsilateral hip arthrodesis procedure is a contraindication to KA [6].

Several surgical techniques for KA have been introduced, such as intramedullary nailing (IMN), external fixation (EF), and compression plating (CP). IMN may confer more rigid fixation and can be performed in a single operation with early mobilization. EF can also be performed in the presence of an active PJI and provides limb lengthening; however, EF requires delayed weight bearing [7]. Good results have also been achieved with CP, but these results depend on sufficient bone stock, and CP involves extensive soft tissue exposure [8]. Which surgical technique for KA is most effective at eradicating infection while resulting in fewer complications and limb salvage is still under debate. Therefore, the aim of this systematic review was to investigate the outcomes of KA after PJI of the knee. Differences in clinical outcomes and complication rates among patients treated using IMN, EF, and CP were compared.

## 2. Materials and Methods

### 2.1. Search Strategy

A systematic review of the published literature was conducted and reported according to the Preferred Reporting Items for Systematic Reviews and Meta-Analyses (PRISMA) statement [9]. The PubMed, MEDLINE, Scopus, and Cochrane Central databases were searched in March 2024. The search terms used to retrieve relevant articles were “arthrodesis”, “knee”, “arthroplasty”, “infection”, “periprosthetic knee infection”, “results”, “outcome”, “eradication”, and “complications”. The study protocol was registered in PROSPERO (CRD42024516659). Two authors (MM and AZ) independently screened the titles and abstracts to identify articles for inclusion, and they contacted a third senior author (OG) in cases of major discrepancies. The reference list of each included article, as well as the available gray literature at our institution, was screened for potential additional articles [1].

### 2.2. Inclusion Criteria and Study Selection

The following inclusion criteria were applied during title, abstract, and full-text screening according to the PICO [10] format: (1) Population: patients who underwent KA for PJI of the knee; (2) Intervention: studies on IMN, EF, and CP reporting >5 surgically treated cases; (3) Comparator: all studies were included irrespective of the presence or absence of comparator or control groups; and (4) Outcome: articles written in English reporting outcomes and/or complications of IMN, EF, and CP with a minimum mean follow-up of 12 months. Other reviews, case reports, articles without outcomes or results, cadaveric or biomechanical studies, technical notes, editorials, letters to the editor, and expert opinions were excluded from the analysis but considered for the Discussion section.

### 2.3. Data Extraction and Quality Assessment

Two surgeons (MM and AZ) examined the included studies and extracted the data. The first author, journal name, year of publication, type of surgery, and patient demographics were recorded for each article [11]. The data extracted for quantitative analysis included the visual analog scale (VAS) score for pain, the time from TKA to KA, and the number and types of complications. The fusion rate, reoperations, and amputations were also analyzed. Reoperations were defined as interventions requiring any return to the operating room for any reason, excluding conversion to amputation [10,11]. A methodological quality assessment was independently conducted by 3 authors (MM, EC, and AZ) using the Modified Newcastle–Ottawa Quality Assessment Scale [12]. The discrepancies were resolved by consulting a senior reviewer with over 25 years of experience in knee surgery (OG). Details of the quality assessment are shown in Table 1.

### 2.4. Statistical Analysis

The quantitative data were organized for statistical analysis; all the data were collected, measured, and reported with 1-decimal accuracy. KA cases were divided into 3 groups: IMN, EF, and CP. Weighted means and standard deviations were calculated for data concerning demographic characteristics and outcomes. When standard deviations were not directly provided, they were calculated with the equation (max range−min range/4) to allow for statistical aggregation [30]. The weighted mean and standard deviation comparisons were performed using unpaired t tests, and 2 × 2 contingency tables were used to compare proportions. All tests were performed with SPSS Statistics software (version 25.0; IBM Corp., Armonk, NY, USA) and GraphPad Prism (version 7.0; GraphPad Software Inc., San Diego, CA, USA). The 95% confidence intervals were calculated, and a *p* value less than 0.05 was considered to indicate statistical significance.

## 3. Results

In total, 675 relevant articles were identified through the initial search, 325 abstracts were screened, and 70 full-text articles were assessed for eligibility based on our inclusion criteria. This resulted in 23 studies that were eligible for systematic review (Figure 1).

A total of 888 patients were initially identified, and 787 were treated and evaluated, of whom, 601 (76.4%), 166 (21%), and 19 (2.4%) underwent IMN, EF, and CP, respectively. The baseline characteristics of these studies are summarized in Table 2.

Overall, 52.3% of the patients were female. The frequency-weighted mean age at the time of the operation was 67.9 ± 9 years, and the frequency-weighted mean follow-up was 60.4 ± 54.1 months.

The rates of comorbidities and smoking habits are reported in Table 3. The prevalence of diabetes was 30%.

The indications for KA were primary and recurrent PJI. Overall, the preoperative VAS score was evaluated in three studies [18,20,28] involving a total of 91 patients, with a mean value of 5.1 ± 3.5. In eight studies [3,4,16,19,24,25,28], the postoperative VAS score was evaluated in 217 patients, with a mean value of 2.7 ± 2.2, and a statistically significant difference was found (*p* < 0.001).

In six studies [14,25,26,27,31,32], the time between TKA and KA was reported and a frequency-weighted mean time of 39.7 ± 44 months was found.

**Table 2 healthcare-12-00804-t002:** Characteristics of included studies.

Author	Journal	Year of Publication	Years of Study	Type of Surgery	Patient Demographics											
					Number of Patients (*N*)	Sex (*N*)		Age (years)			Time between TKA and Treatment (months)			FU (months)		
						M	F	Mean	SD	Range	Mean	SD	Range	Mean	SD	Range
**Aparicio et al. [31]**	*Indian Journal of Orthopaedics*	2020	2001–2019	IMN	45	13	32	72	8.3	57–90	55	57	12–240	102	45.5	24–206
**Balci et al. [14]**	*Journal of Knee Surgery*	2015	1999–2012	EF	17	14	3	67	16.6	29–93	6.8	2.4	5.6–8	62.96	34	24–160
**Brown et al. [33]**	*Journal of the American Academy of Orthopaedic Surgeons*	2020	2004–2013	IMN	17	7	11	65	12.8	32–83	NA	NA	NA	50	37	2–150
**Corona et al. [16]**	*European Journal of Orthopaedic Surgery & Traumatology*	2020	2014–2018	EF	29	13	16	77.96	7.7	39–88	NA	NA	NA	47.1	17	12–82.8
**Faure et al. [17]**	*Orthopaedic & Traumatology: Surgery & Research*	2021	2005–2005	IMN	31	12	19	67	12	48–80	NA	NA	NA	158	6.3	138–163
**Friedrich et al. [4]**	*The Knee*	2017	2008–2014	IMN	32	NA	NA	70.2	11.5	43–89	NA	NA	NA	31	15.5	12–74
**Galluser et al. [32]**	*European Journal of Orthopaedic Surgery & Traumatology*	2015	2004–2012	IMN	12	NA	NA	67	11.3	42–87	2	1.2	0.5–5.3	33	31.5	6–132
**Gathen et al. [3]**	*Archives of Orthopaedic and Trauma Surgery*	2018	2008–2014	IMN	36	16	20	69.9	10.4	NA	NA	NA	NA	34.6	17.7	NA
**Gramlich et al. [18]**	*Archives of Orthopaedic and Trauma Surgery*	2021	2010–2017	IMN	52	23	29	73.9	1.6	49–97	NA	NA	NA	81.6	46.5	6–192
**Hawi et al. [19]**	*The Bone & Joint Journal*	2015	2002–2012	IMN	27	17	10	68.8	8.8	52–87	NA	NA	NA	67.1	29.8	24–143
**Iacono et al. [20]**	*HSS Journal*	2013	2001–2009	IMN	22	15	19	69.3	8	53–85	NA	NA	NA	34.4	1	13–17
				EF	12			68.5	7.3	55–84	NA	NA	NA	93.2	7.3	82–111
**Putman et al. [21]**	*Orthopaedic & Traumatology: Surgery & Research*	2013	2005–2008	IMN	31	NA	NA	67	12	78–80	NA	NA	NA	50	22	28–90
**Razii et al. [8]**	*European Journal of Orthopaedic Surgery & Traumatology*	2016	2003–2014	IMN	12	9	3	67	13.3	35–88	NA	NA	NA	48.5	27.8	9–120
**Robinson et al. [22]**	*Journal of the American Academy of Orthopaedic Surgeons*	2018	2002–2014	IMN/EF	21	9	14	63.7	NA	NA	18.2	NA	NA	4.4	126	12–138
**Rohner et al. [23]**	*The Journal of Bone and Joint Surgery*	2015	1997–2013	IMN	26	8	18	68	10	48–88	NA	NA	NA	NA	NA	NA
**Stavrakis et al. [15]**	*Arthroplasty Today*	2022	1998–2019	IMN	81	39	42	67	7.9	45–84	25	NA	NA	52	NA	NA
**Suda et al. [24]**	*International Orthopaedics*	2021	2014–2018	IMN/EF/CP	152	81	78	63.6	19.5	12–90	NA	NA	NA	36	9.25	12–49
**Troulliez et al. [25]**	*Orthopaedic & Traumatology: Surgery & Research*	2021	2003–2019	IMN	23	7	16	68	11.1	53–81	48	6.3	24–87	116.34	56.8	13.2–171.96
**Vivacqua et al. [13]**	*Revista brasilera de Ortopedia*	2021	2010–2016	EF	18	9	9	NA	NA	NA	NA	NA	NA	44.4	NA	NA
**Watanabe et al. [26]**	*Modern Rheumatology*	2014	2005–2007	EF	8	1	7	72.9	4.3	63–80	23.3	16.25	5–70	39	2.25	36–45
**Wilding et al. [27]**	*The Journal of Arthroplasty*	2016	2008–2014	IMN	8	2	6	73.8	3.9	67–82.5	78	36	6–150	16	7.5	5–35
**Yeung et al. [28]**	*The Journal of Arthroplasty*	2020	2000–2016	IMN/EF/CP	51	23	28	65	4.5	55–73	NA	NA	NA	78	99	43–142
**Zajonz et al. [29]**	*Der Orthopäde*	2021	2010–2016	IMN	18	NA	NA	76.6	5	60.6–80.5	NA	NA	NA	51	18	10–82
				CP	7	NA	NA	60.6	6.3	55–80.5	NA	NA	NA	28	10.5	2–44

FU follow-up, SD standard deviation, TKA total knee arthroplasty, VAS visual analogue scale, IMN intramedullary nailing, EF external fixation, BPEMF biplanar extramedullary fixation, CP compression plating, NA not available.

**Table 3 healthcare-12-00804-t003:** Prevalence of comorbidities and smoke habits.

Comorbidities	N. of Studies	N. of Patients	Prevalence
Diabetes	11	138	30%
Hypertension	7	209	63%
Cardiac Disease	6	44	13.8%
Rheumatoid Arthritis	9	45	16.9%
Smoke Habits	3	105	57.1%

One-stage KA was performed in 18.7% of patients, while two-stage KA was performed in 81.3% of patients. Aggregated data on the type of implant used for the IMN procedures included the Link™ Nail (Link, Hamburg, Germany) [19,20,25], Link Endomodel™ (Link, Boves, France) [17,21], Wichita Fusion Nail^®^ [32], KAM-TITAN^®^ (Peter Brehm GmbH, Weisendorf, Germany) [18], OSS Modular Arthrodesis System (Zimmer Biomet, Warsaw, IN, USA) [15], long Küntscher nail [8,31], RS Arthrodesis Implant (Implantcast, Buxtehude, Germany) [8,27], T2 fusion nail (Stryker, Kalamazoo, MI, USA) [31,33], and Modular Arthrodesis System (Peter Brehm GmbH, Weisendorf, Germany) [3]. Different systems were used for EF procedures, such as the limb reconstruction system (LRS, Orthofix, Bussolengo, Italy) [14,16], the Monotube^®^ external fixator (Stryker, Kalamazoo, MI, USA) [26], and the Hoffmann II (Stryker, Kalamazoo, MI, USA) [20]. Studies that used CPs did not specify the type of implant.

### 3.1. Microbiology

The causative pathogens are reported in Table 4.

Data for perioperative cultured pathogens were reported for 857 cultures, and the most common causative bacteria were coagulase-negative Staphylococcus (CNS) (7%), Staphylococcus epidermidis (6.1%), and methicillin-resistant Staphylococcus aureus (MRSA) (5.8%). Cultures not specified amounted to 45.5% of the total. A total of 321 patients had a single causative pathogen, while 172 patients had multiple causative pathogens.

### 3.2. Complications

The wound dehiscence rates were 6%, 16.3%, and 5.3% after IMN, EF, and CP, respectively; EF had a greater wound dehiscence rate than did IMN (*p* < 0.001) (Table 5).

The fracture rates were 3.4%, 6.6%, and 5.3% after IMN, EF, and CP, respectively, and no statistically significant differences were found among the groups. The transient nerve injury rates were 0.2%, 0%, and 0% after IMN, EF, and CP, respectively, and no statistically significant differences were found among the groups. The permanent nerve injury rates were 0.7%, 2.4%, and 0% after IMN, EF, and CP, respectively, and no statistically significant differences were found among the groups. The aseptic loosening rates were 0.8%, 0%, and 0% after IMN, EF, and CP, respectively, and no statistically significant differences were found among the groups. The long-term suppressive antibiotic treatment rates were 2.8%, 0%, and 0% after IMN, EF, and CP, respectively; IMN resulted in a higher rate of long-term suppressive antibiotic treatment than did EF (*p* = 0.03). The reinfection rates were 14.6%, 15.1%, and 10.5% after IMN, EF, and CP, respectively, and no statistically significant differences were found among the groups.

### 3.3. Fusion, Reoperation and Amputation Rates

Fusion was defined as bony trabeculae traversing from the tibia to the femur in at least two radiographic projections. It was reported after IMN in nine studies [8,15,21,22,24,29,31,32,33] and occurred in 192 out of 267 patients (71.9%). In the EF group, fusion was reported in seven studies [13,14,16,20,24,26,29] and occurred in 130 out of 165 (78.8%) patients. Fusion after CP was reported in two studies [13,29] and occurred in 12 out of 13 patients (92.3%). No statistically significant differences were found among the groups.

Reoperations occurred in 11%, 9%, and 0% of patients after IMN, EF, and CP, respectively, and no statistically significant differences were found among the groups.

Conversion to amputation occurred in 4.3%, 5%, and 15.8% of patients after IMN, EF, and CP, respectively; CP resulted in a higher amputation rate than did EF (*p* = 0.03), and no statistically significant difference was found between IMN and EF.

## 4. Discussion

The aim of this systematic review was to investigate the outcomes of KA after PJI of the knee. We noted that the most common causative pathogens causing PJI treated with KA were *Staphylococcus* species, and this surgical procedure confirmed the effectiveness of treating the pain associated with PJI. The most common procedure used to achieve KA was IMN, and 81.3% of patients underwent a two-stage KA procedure. Comparing IMN with EF, a higher rate of long-term suppressive antibiotic use was reported in patients who underwent IMN, while the EF group reported a higher rate of wound dehiscence; however, no differences in terms of fusion, reinfection, or conversion to amputation were reported. CP is rarely used, and the high conversion-to-amputation rate represents an important limitation of this technique.

This study updates the current evidence of previous systematic reviews by including the largest number of studies on the subject, with a total of 23 studies evaluating a greater number of endpoints. While other systematic reviews exist in the literature on this topic, they are mostly descriptive in nature [6] and report only part of the available data [7], which increases the risk of bias. Recent studies have been published with new techniques and devices to achieve KA, and the current study allows a clearer understanding of the comparative efficacy of these treatment modalities that may be of interest for patients, clinicians, researchers, and policymakers in the orthopedic field and may guide evidence-based decision making, ultimately leading to improved patient outcomes.

The key principles for KA after PJI include preoperative patient optimization, infection control, optimal knee fusion position, maximum bone contact at the fusion site, and achieving a desirable leg length [6]. In the current review, the patients’ mean age was >65 years and their diabetes and rheumatoid arthritis rates and demographic characteristics were in line with previously identified patient-related risk factors for persistent PJI of the knee [34]. Staphylococcus species are the most common pathogens causing PJI [35]; in the current study, the prevalence of PJI caused by CNS, Staphylococcus epidermidis, and MRSA accounted for almost 20%of the total. Moreover, in half of the cases, an infection with multiple causative pathogens was noted. The data concur with those reported by Vasso et al. [5], who determined the incidence and predictors of failure in patients treated with two-stage reimplantation for PJI and found that the number of highly virulent bacterial infections increased due to inappropriate antibiotic treatment strategies in community and health care settings [36,37,38]. Other studies have also reported that Pseudomonas aeruginosa infection and PJI caused by multiple causative bacteria were risk factors for amputation [39].

We found that the most common procedure used to achieve KA was IMN, and 81.3% of patients underwent a two-stage KA procedure. Notably, IMN is a surgical technique that is more familiar to surgeons than EF and the use of antibiotic cement-coated interlocking IMNs has been proposed to improve PJI eradication, expanding the indications for single-stage procedures [40]. Contraindications to the IMN technique include severe bone loss and ipsilateral femoral or tibial deformity, whereas leg length discrepancies cannot be corrected without future elongation surgery. Five to seven degrees of anatomical femoral valgus has been suggested to be the optimal alignment in the coronal plane; considering the leg length, some authors have suggested achieving full knee extension with KA, whereas others have recommended 10° to 15° of flexion to shorten the leg length by 1.5 cm compared with the contralateral leg length, thereby improving the gait speed and sitting position [41]. Different types of IMN procedures have been used, including nonmodular, modular, and long interlocking nails. Modular nails have the advantage of two separate components that fit better into the canals and they are connected with a coupler device, while short nails should also be used in the presence of bone deformity or ipsilateral hip arthroplasty. We reported that the fusion rate after IMN was 72%, which was similar to the rate reported after EF (78.8%). In contrast, White et al. [7] reported a higher fusion rate in the IMN group, but they noted publication bias and heterogeneity in terms of the participants and settings between the IMN and EF groups. No difference in the rate of conversion to amputation was found between the IMN and EF groups, and the data concurred with those already reported by White et al. [7]. Postoperative infection rates were also similar to those previously reported and ranged from 10% to 15%, regardless of the technique adopted. These data concur with those reported in a previous meta-analysis [36] that showed a reinfection rate of 13% after the IMN technique.

EF represents an effective option in cases of severe bone loss, poor soft-tissue coverage, and contraindications for internal fixation. In addition, EF can be used to treat active PJI in a single-stage procedure, limiting femoral and tibial intramedullary dissemination. In the current review, EF was used in 21% of the patients with different configurations. The circular fixator allows for the best biomechanical stability, limb lengthening, and deformity correction [42], but a uniplanar or biplanar fixator is often preferred because it is a less demanding procedure. In the current study, an overall fusion rate of 73% was noted with the EF technique. However, Oostenbroek et al. [43] analyzed the EF technique using an Ilizarov ring fixator in 15 patients and reported that the fusion rate can be up to 90%.

CP can be used if the bone loss is minimal and there is no soft-tissue coverage problem. CP provides rigid fixation, and a fusion rate greater than 90% was reported in the current study. However, this technique is rarely used to perform KA. An increased surgical dissection and inability to fully bear weight after the procedure has been previously reported [44], whereas a high rate of conversion to amputation was found in the current study, thus representing important limitations of this technique.

The findings of this study should be interpreted while considering several limitations. First, only studies in the English language were included, potentially contributing to publication bias; moreover, although four major literature databases were used for this search, we cannot exclude the possibility that additional articles could have been found using other databases. Second, heterogeneity in terms of sample size was found between the included studies. Third, a high degree of heterogeneity among the articles included was noted in terms of the type of implant in each group and it was not possible to compare outcomes according to the implant used. Fourth, it was not possible to perform a comparison in terms of time to fusion since the relative data were not adequate for a statistical aggregation. Finally, we noted heterogeneity of the mean follow-ups of the included studies; it is likely that complication rates are affected by the length of the evaluation times and these outcomes could also be potentially different among the procedures if a specific and longer follow-up time was applied. Clinicians should always consider the differences in patient characteristics that may favor certain treatment options, and appropriate patient selection is critically important to maximize outcomes.

## 5. Conclusions

*Staphylococcus* species are the most common causative microorganisms of PJI cases treated by KA, and this procedure is an effective surgical technique for treating pain symptoms associated with PJI. The intramedullary nail technique and a two-stage KA procedure are the most common procedures used. A higher rate of long-term suppressive antibiotics was reported in the intramedullary nail group, while the external fixator group reported a higher rate of wound dehiscence; however, no differences in terms of fusion, reinfection, or conversion to amputation were reported between the intramedullary nail group and the external fixator group. Compression plating is rarely used, and the high conversion-to-amputation rate is an important limitation of this technique. Future randomized controlled trials that include patients undergoing KA using the same implant should be conducted to confirm these findings.

## Figures and Tables

**Figure 1 healthcare-12-00804-f001:**
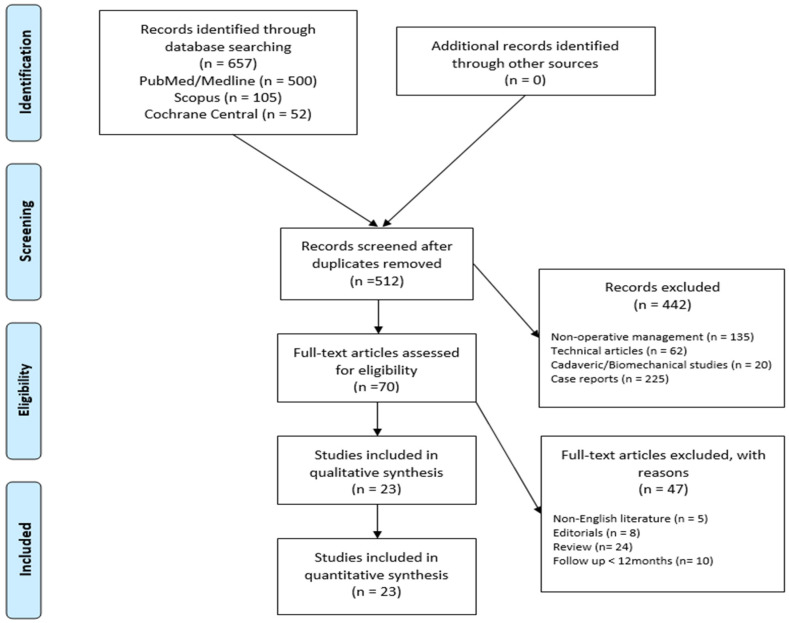
Preferred Reporting Items for Systematic Review and Meta-Analysis (PRISMA) flowchart for the searching and identification of included studies.

**Table 1 healthcare-12-00804-t001:** Newcastle–Ottawa Scale.

Study Author (year)	Criteria								Total	Quality
	1	2	3	4	5	6	7	8		
Aparicio et al. (2020) [13]	1	0	1	1	2	1	1	1	8	High
Balci et al. (2015) [14]	1	0	1	1	2	1	1	1	8	High
Brown et al. (2020) [15]	1	0	1	1	2	1	1	1	8	High
Corona et al. (2020) [16]	1	1	1	1	2	1	1	1	9	High
Faure et al. (2021) [17]	1	0	1	1	2	1	1	1	8	High
Friedrich et al. (2017) [4]	1	0	1	1	2	1	1	1	8	High
Galluser et al. (2015) [14]	1	0	1	1	2	1	1	1	8	High
Gathen et al. (2018) [3]	1	1	1	1	2	1	1	1	9	High
Gramlich et al. (2021) [18]	1	1	1	1	2	1	1	1	9	High
Hawi et al. (2015) [19]	1	0	1	1	2	1	1	1	8	High
Iacono et al. (2013) [20]	1	1	1	1	2	1	1	1	9	High
Putman et al. (2013) [21]	1	0	1	1	2	1	1	1	8	High
Razii et al. (2016) [8]	1	0	1	1	2	1	1	1	8	High
Robinson et al. (2018) [22]	1	0	1	1	2	1	1	1	8	High
Rohner et al. (2015) [23]	1	1	1	1	2	1	1	1	9	High
Stavrakis et al. (2022) [15]	1	0	1	1	2	1	1	1	8	High
Suda et al. (2021) [24]	1	1	1	1	2	1	1	1	9	High
Trouillez et al. (2021) [25]	1	1	1	1	2	1	1	1	9	High
Vivacqua et al. (2021) [13]	1	0	1	1	2	1	1	1	8	High
Watanabe et al. (2014) [26]	1	0	1	1	2	1	1	1	8	High
Wilding et al. (2016) [27]	1	0	1	1	2	1	0	1	7	High
Yeung et al. (2020) [28]	1	0	1	1	2	1	1	1	8	High
Zajonz et al. (2021) [29]	1	1	1	1	2	1	1	1	9	High

Based on the total score, quality was classified as “low” (0–3), “moderate” (4–6), and “high” (7–9). Criterion number (in bold): 1, representativeness of the exposed cohort; 2, selection of the nonexposed cohort; 3, ascertainment of exposure; 4, demonstration that outcome of interest was not present at start of study; 5, comparability of cohorts on the basis of the design or analysis; 6, assessment of outcome; 7, was follow-up long enough for outcomes to occur?; 8, adequacy of follow up of cohorts. Each study was awarded a maximum of one or two points for each numbered item within categories, based on the Modified Newcastle–Ottawa Scale rules.

**Table 4 healthcare-12-00804-t004:** Causative pathogens in periprosthetic knee infections undergoing knee arthrodesis.

Microorganism	No.	%
*Not specified*	390	45.5
*Negative cultures/no bacterial growth*	65	7.6
*Coagulase-negative Staphylococcus*	60	7
*Staphylococcus Epidermidis*	52	6.1
*Methicillin-Resistant Staphylococcus aureus*	50	5.8
*Methicillin-Sensitive Staphylococcus aureus*	45	5.2
*Staphylococcus Aureus non specified*	35	4
*Pseudomonas Aeruginosa*	23	2.7
*Enterococcus faecalis*	23	2.2
*Streptococcus* spp.	18	2.1
*Polymicrobial not otherwise specified*	16	1.9
*Escherichia Coli*	15	1.7
*Enterobacter Cloacae*	6	0.7
*Candida*	5	0.6
*Vancomycin-Resistant Enterococcus*	4	0.5
*Enterococcus* spp.	4	0.5
*Proteus Mirabilis*	4	0.5
*Propionibacterium Acnes*	3	0.3
*Streptococcus Agalactiae*	3	0.3
*Fusobacterium* spp.	3	0.3
*Staphylococcus Capitis*	3	0.3
*Serratia marcescens*	3	0.3
*Methicillin-Resistant Staphylococcus epidermidis*	2	0.2
*Clostridium*	2	0.2
*Corynebacterium non specified*	2	0.2
*Klebsiella Pneumoniae*	2	0.2
*Prevotella bivia*	2	0.2
*Stafilococcus Warneri*	2	0.2
*Streptococco Dysagalactie*	2	0.2
*Enterococcus Faecium*	2	0.2
*Streptococcus Viridans*	2	0.2
*Methicillin-Sensitive Staphylococcus epidermidis*	1	0.1
*Enterobacter* spp.	1	0.1
*Corynebacterium Amicolatum*	1	0.1
*Staphylococcus Haemoliticus*	1	0.1
*Staphylococcus Hominis*	1	0.1
*Mycobacterium*	1	0.1
*Klebsiella*	1	0.1
*Acinetobacter Baumanii*	1	0.1
*Morganelli Morgana*	1	0.1
Total	857	

**Table 5 healthcare-12-00804-t005:** Comparison of complications among IMN, EF, and CP groups.

Complications	IMN vs. CP	IMN vs. EF	EF vs. CP
Wound dehiscence	*p* = 1	***p* < 0.001**	*p* = 0.3
Fractures	*p* = 0.4	*p* = 0.07	*p* = 1
Transient nerve injury	*p* = 1	*p* = 1	*p* = 1
Permanent nerve injury	*p* = 1	*p* = 0.07	*p* = 1
Aseptic loosening	*p* = 1	*p* = 0.59	*p* = 1
Long-term suppressive antibiotics treatment	*p* = 1	***p* = 0.03**	*p* = 1
Reinfection	*p* = 1	*p* = 0.9	*p* = 1
Reoperation	*p* = 0.24	*p* = 0.56	*p* = 0.37
Conversion to amputation	*p* = 0.053	*p* = 0.65	***p* = 0.03**

IMN intramedullary nailing, CP compression plating, EF external fixation. Results with *p* value < 0.05 are in bold.

## Data Availability

Not applicable.

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
