# Peer review of "Knee Arthrodesis for Periprosthetic Knee Infection: Fusion Rate, Complications, and Limb Salvage—A Systematic Review"

_healthcare, 2024, doi:10.3390/healthcare12070804_

Round 1

Reviewer 1 Report

Comments and Suggestions for Authors

I read 'Knee arthrodesis for periprosthetic knee infection: Fusion rate, complications, and limb salvage - A systematic review' with great interest. In this manuscript, the authors aimed to evaluate the outcomes of knee arthrodesis (KA) after periprosthetic joint infection (PJI) of the knee. Overall, the paper contributes to the understanding of evidence on the outcomes of KA after PJI, including a larger number of studies and endpoints, reducing the risk of bias.

I would like to pose several questions for further clarity:

  1. Could you provide a more detailed explanation of what makes this study innovative or how it contributes new insights to the field?
  2. Why is CP rarely used for KA?
  3. What are the differences in outcomes between IMN and EF for managing PJI with KA?
  4. What were the three surgical techniques compared in this study?
  5. Which pathogen was most commonly associated with periprosthetic joint infection?
  6. What percentage of patients achieved fusion after each surgical technique?
  7. Did any statistically significant differences exist in reinfection rates among the three techniques?
  8. Which technique had the highest rate of conversion to amputation?
  9. What percentage of patients achieved fusion after each procedure?
  10. Did any of the surgical techniques have a statistically significant difference in reinfection rates?
  11. How do the methodological limitations affect the interpretation of results?
  12. Discussion is rather weak. Please compare the results of this study with the results of other studies for a more in-depth discussion
  13. Could you provide more data or discussion on the importance of healthcare professionals worldwide recognizing the importance of the understanding of the outcomes and complications associated with KA after PJI, providing valuable insights for clinical decision-making.
Comments on the Quality of English Language

Minor editing of English language required

Author Response

Reviewer #1

I read 'Knee arthrodesis for periprosthetic knee infection: Fusion rate, complications, and limb salvage - A systematic review' with great interest. In this manuscript, the authors aimed to evaluate the outcomes of knee arthrodesis (KA) after periprosthetic joint infection (PJI) of the knee. Overall, the paper contributes to the understanding of evidence on the outcomes of KA after PJI, including a larger number of studies and endpoints, reducing the risk of bias.:

A: Thanks for your comment. We revised the text point-to-point according to your valuable suggestions.

Could you provide a more detailed explanation of what makes this study innovative or how it  contributes new insights to the field?

A: Thanks for your comment. We revised the text according to your comment.

This study updates the current evidence of previous systematic reviews by including the largest number of studies on the subject, with a total of 23 studies evaluating a greater number of endpoints. While other systematic reviews exist in the literature on this topic, they are mostly descriptive in nature[6] and report only part of the available data[7], which increases the risk of bias. Recent studies have been published with new techniques and devices to achieve KA and the current study is a clearer understanding of the comparative efficacy of these treatment modalities that may be of interest for patients, clinicians, researchers, and policy-makers in the orthopaedic field and may guide evidence-based decision-making, ultimately leading to improved patient outcomes.

(page 11, lines 284-292)

Why is CP rarely used for KA?

A: Thanks for your comment. We revised the text according to your comment.

CP can be used if the bone loss is minimal and there is no soft-tissue coverage problem. CP provides rigid fixation, and a fusion rate greater than 90% was reported in the current study. However, this technique is rarely used to perform KA. It has been reported inability to fully bear weight after the procedure and an increased surgical dissection[43]; a high rate of conversion to amputation has been reported in the current study, thus representing an important limitations of this technique.

(page 12, lines 381-386)

What are the differences in outcomes between IMN and EF for managing PJI with KA?

A: Thanks for your comment.

The included studies did not provide comparable data in terms of outcomes. We provided these aspects in the limitations section according to your comment.

Third, a high degree of heterogeneity among the articles included was noted in terms of the type of implant in each group, and it was not possible to compare functional outcomes according to the implant used.

(page 13, lines 411-413)

What were the three surgical techniques compared in this study?

A: Thanks for your comment. We explain the three surgical techniques compared in the introduction.

Several surgical techniques for KA have been introduced, such as intramedullary nailing (IMN), external fixation (EF) and compression plating (CP). […]Therefore, the aim of this systematic review was to investigate the outcomes of KA after PJI of the knee. Differences in clinical outcomes and complication rates among patients treated by IMN, EF, and CP were compared.

(page 2, lines 52-53, 60-62)

Which pathogen was most commonly associated with periprosthetic joint infection?

A: Thanks for your comment. An interesting study reported that Staphylococcus species were the most common pathogens causing PJI. This data concurs with those reported in our systematic review. We revised the text providing this data according to your comment.

Staphylococcus species are the most common pathogens to cause PJI [34] and in the current study the prevalence of PJI caused by CNS, Staphylococcus epidermidis, and MRSA accounted for almost 20%; moreover, in half of the cases, an infection with multiple causative pathogens was noted.

(page 12, lines 337-340).

What percentage of patients achieved fusion after each surgical technique?

A: Thanks for your comment. We reported the fusion rate in the results section as follows.

Fusion was defined as bony trabeculae traversing from the tibia to the femur in at least two radiographic projections; it was reported after IMN in 9 studies[8,19,22,14,28-32] and occurred in 192 out of 267 patients (71.9%). In the EF group, fusion was reported in 7 studies[16,17,19,21,23,25,32] and occurred in 130 out of 165 (78.8%) patients. Fusion after CP was reported in 2 studies[21,32] and occurred in 12 out of 13 patients (92.3%). No statistically significant differences were found among the groups.

(page 11, lines 261-266)

Did any statistically significant differences exist in reinfection rates among the three techniques?

A: Thanks for your comment. We reported the reinfection rate in the results section and Table 5 as follows.

No statistically significant differences were reported between the IMN and CP techniques (P=1), IMN and EF techniques (P=0.9), and EF and CP techniques (P=1).

(pages 10-11, table 5)

Which technique had the highest rate of conversion to amputation?

A: Thanks for your comment. We reported the conversion to amputation rate in the results section as follows.

Conversion to amputation occurred in 4.3%, 5%, and 15.8% of patients after IMN, EF and CP, respectively; CP resulted in a higher amputation rate than did EF (p = 0.03), and no statistically significant difference was found between IMN and EF.

(page 11, lines 269-271).

What percentage of patients achieved fusion after each procedure?

A: Thanks for your comment. We reported the fusion rate in the results section as follows.

Fusion was defined as bony trabeculae traversing from the tibia to the femur in at least two radiographic projections; it was reported after IMN in 9 studies [8,19,22,14,28-32] and occurred in 192 out of 267 patients (71.9%). In the EF group, fusion was reported in 7 studies [16,17,19,21,23,25,32] and occurred in 130 out of 165 (78.8%) patients. Fusion after CP was reported in 2 studies [21,32] and occurred in 12 out of 13 patients (92.3%). No statistically significant differences were found among the groups.

(page 11, lines 261-266)

Did any of the surgical techniques have a statistically significant difference in reinfection rates?

A: Thanks for your comment. We reported the reinfection rate in the results section as follows.

No statistically significant differences were reported between the IMN and CP techniques (P=1), IMN and EF techniques (P=0.9), and EF and CP techniques (P=1). Reinfection rates were 14.6%, 15.1%, and 10.5% after IMN, EF and CP, respectively, and no statistically significant differences were found among the groups.

(pages 10-11, lines 257-259 and table 5)

How do the methodological limitations affect the interpretation of results?

A: Thanks for your comment. We reported the methodological limitations in the discussion section.

Second, heterogeneity in terms of sample size was found between the included studies. Third, a high degree of heterogeneity among the articles included was noted in terms of the type of implant in each group, and it was not possible to compare outcomes according to the implant used. Fourth, it was not possible to perform a comparison in terms of time to fusion since the relative data were not adequate for a statistical aggregation. Finally, we noted heterogeneity of the mean follow-ups of the included studies it is likely that complication rates are affected by the length of the evaluation times and these outcomes could also be potentially different among the procedures if a specific and longer follow-up time was applied. Clinicians should always consider the differences in patient characteristics that may favor a certain treatment option and appropriate patient selection is critically important to maximize outcomes.

(page 13, lines 434-444)

Discussion is rather weak. Please compare the results of this study with the results of other studies for a more in-depth discussion

A: Thanks for your comment. We improved our discussion in the text accordingly.

Postoperative infection rates were also similar to those previously reported and ranged from 10% to 15%, regardless of the technique adopted. These data concur with those reported in a previous meta-analysis [35] that showed a reinfection rate of 13% after IMN technique.

(page 12, lines 368-369)

In the current study, an overall fusion rate of 73% was noted with the EF technique. However, Oostenbroek et al.[42] analysed the EF technique using an Ilizarov ring fixator in 15 patients and reported that the fusion rate can be up to 90%.

(page 12, lines 376-379)

Could you provide more data or discussion on the importance of healthcare professionals worldwide recognizing the importance of the understanding of the outcomes and complications associated with KA after PJI, providing valuable insights for clinical decision-making.

A: Thanks for your comment. We revised the text according to your comment.

This study updates the current evidence of previous systematic reviews by including the largest number of studies on the subject, with a total of 23 studies evaluating a greater number of endpoints. While other systematic reviews exist in the literature on this topic, they are mostly descriptive in nature[6] and report only part of the available data[7], which increases the risk of bias. Recent studies have been published with new techniques and devices to achieve KA and the current study is a clearer understanding of the comparative efficacy of these treatment modalities that may be of interest for patients, clinicians, researchers, and policy-makers in the orthopaedic field and may guide evidence-based decision-making, ultimately leading to improved patient outcomes.

(page 11, lines 283-291)

Reviewer 2 Report

Comments and Suggestions for Authors

This study is a systematic review that examined knee arthrodesis (KA) as a treatment for periprosthetic joint infection (PJI). Staphylococcus species were identified as the most common pathogens in PJI cases treated with KA, with intramedullary nailing (IMN) being the most frequently used procedure. The review included 23 studies and highlighted the effectiveness of KA in managing PJI-associated pain. IMN was associated with higher long-term antibiotic use, while external fixation (EF) had a higher rate of wound dehiscence. However, no significant differences were found in terms of fusion, reinfection, or conversion to amputation between IMN and EF. Compression plating (CP) was rarely used, with a high conversion-to-amputation rate.

Strengths:

  • The study followed the PRISMA guidelines for systematic reviews, ensuring a systematic and transparent approach to study selection and data extraction.
  • The study conducted a thorough search across multiple databases, minimizing the risk of missing relevant studies.
  • By comparing outcomes between different KA procedures, the study provides valuable insights into the effectiveness and safety of each approach.
  • Knee arthrodesis is a significant procedure in the management of PJI, and the study's findings have direct implications for clinical practice, potentially guiding treatment decisions.

Limitations:

  • Variability in study designs, patient populations, and outcome measures among the included studies may limit the generalizability of the findings.
  • Some outcomes, such as time to fusion, were not consistently reported across studies, limiting the ability to draw firm conclusions about these outcomes.

Detailed Comments:

1.     The introduction briefly mentions the different surgical techniques for KA (IMN, EF, CP) and their potential advantages and disadvantages. However, it could be more informative by discussing specific studies or evidence supporting these statements. Additionally, the discussion on contraindications to KA could be expanded to provide more context and reasoning.

2.     While the introduction covers a wide range of topics related to PJI and KA, the presentation could be more concise and focused. Some sections feel slightly disjointed, and the introduction could benefit from clearer transitions between ideas.

3.     The study uses the Modified Newcastle–Ottawa Quality Assessment Scale to assess the quality of included studies, which is a commonly used tool in systematic reviews. However, the study could provide more detail on how the quality assessment was conducted (e.g., criteria used for scoring) to ensure transparency.

4.     Ensure that Figure 1 is renamed with a descriptive title other than just "PRISMA."

5.     Upon reviewing the references, it appears that you have cited MORE THAN TEN references from your own previous work. It is essential to note that self-citation without a valid scientific reason is considered unethical. Therefore, I recommend carefully examining the references and eliminating any instances of unnecessary self-citation to ensure the integrity of the scholarly work.

Author Response

Reviewer #2

This study is a systematic review that examined knee arthrodesis (KA) as a treatment for periprosthetic joint infection (PJI). Staphylococcus species were identified as the most common pathogens in PJI cases treated with KA, with intramedullary nailing (IMN) being the most frequently used procedure. The review included 23 studies and highlighted the effectiveness of KA in managing PJI-associated pain. IMN was associated with higher long-term antibiotic use, while external fixation (EF) had a higher rate of wound dehiscence. However, no significant differences were found in terms of fusion, reinfection, or conversion to amputation between IMN and EF. Compression plating (CP) was rarely used, with a high conversion-to-amputation rate.

Strengths:

The study followed the PRISMA guidelines for systematic reviews, ensuring a systematic and transparent approach to study selection and data extraction.

The study conducted a thorough search across multiple databases, minimizing the risk of missing relevant studies.

By comparing outcomes between different KA procedures, the study provides valuable insights into the effectiveness and safety of each approach.

Knee arthrodesis is a significant procedure in the management of PJI, and the study's findings have direct implications for clinical practice, potentially guiding treatment decisions.

Limitations:

Variability in study designs, patient populations, and outcome measures among the included studies may limit the generalizability of the findings.

Some outcomes, such as time to fusion, were not consistently reported across studies, limiting the ability to draw firm conclusions about these outcomes.

 A: Thanks for your comment. We revised the text point-to-point according to your valuable suggestions.

  1. The introduction briefly mentions the different surgical techniques for KA (IMN, EF, CP) and their potential advantages and disadvantages. However, it could be more informative by discussing specific studies or evidence supporting these statements. Additionally, the discussion on contraindications to KA could be expanded to provide more context and reasoning.

A: Thanks for your comment. We provided the advantages and disadvantages of KA in the introduction section according to your comment.

KA represents a valid alternative to amputation in patients with multiple recurrent PJIs or when the PJI is accompanied by severe bone loss, an unreconstructible extensor mechanism, and poor soft-tissue coverage However, high complication, non-union and infection rates have been reported; in addition the loss of the range of motion of the knee may be deterrent for patients[4]. It should also be considered that KA may provide functional results superior to those of resection arthroplasty and amputation, restoring a stable extremity and weight-bearing ability. A prior contralateral KA, amputation or ipsilateral hip arthrodesis procedure is a contraindication to KA[6].

(pages 1-2, lines 40-51)

We also revised the discussion section providing advantages and disadvantages of the three surgical techniques analysed.

Postoperative infection rates were also similar to those previously reported and ranged from 10% to 15%, regardless of the technique adopted. These data concur with those reported in a previous meta-analysis [35] that showed a reinfection rate of 13% after IMN technique.

EF represents an effective option in cases of severe bone loss, poor soft-tissue coverage and contraindications for internal fixation. In addition, EF can be used to treat active PJI in a single-stage procedure, limiting femoral and tibial intramedullary dissemination. In the current review, EF was used in 21% of the patients with different configurations. The circular fixator allows for the best biomechanical stability, limb lengthening, and deformity correction[41], but a uniplanar or biplanar fixator is often preferred because it is a less demanding procedure. In the current study, an overall fusion rate of 73% was noted with the EF technique. However, Oostenbroek et al.[42] analysed the EF technique using an Ilizarov ring fixator in 15 patients and reported that the fusion rate can be up to 90%.

CP can be used if the bone loss is minimal and there is no soft-tissue coverage problem. CP provides rigid fixation, and a fusion rate greater than 90% was reported in the current study. However, this technique is rarely used to perform KA. It has been reported an increased surgical dissection and inability to fully bear weight after procedure [43]; a high rate of conversion to amputation has been reported in the current study, thus representing an important limitations of this technique.

(pages 12, lines 366-385)

  1. While the introduction covers a wide range of topics related to PJI and KA, the presentation could be more concise and focused. Some sections feel slightly disjointed, and the introduction could benefit from clearer transitions between ideas.

A: Thanks for your comment. We revised the introduction according to your suggestions.

  1. The study uses the Modified Newcastle–Ottawa Quality Assessment Scale to assess the quality of included studies, which is a commonly used tool in systematic reviews. However, the study could provide more detail on how the quality assessment was conducted (e.g., criteria used for scoring) to ensure transparency.

A: Thanks for your comment. We provided data on Modifided Newcatle-Ottawa Quality Assessment Scale in Table 1 caption accordingly.

Based on the total score, quality was classified as “low” (0-3), “moderate” (4-6) and “high” (7-9).Criterion number (in bold): 1, representativeness of the exposed cohort; 2, selection of the nonexposed cohort; 3, ascertainment of exposure; 4, demonstration that outcome of interest was not present at start of study; 5, comparability of cohorts on the basis of the design or analysis; 6, assessment of outcome; 7, was follow-up long enough for outcomes to occur?; 8, adequacy of follow up of cohorts. Each study was awarded a maximum of one or two points for each numbered item within categories, based on the Modified Newcastle-Ottawa scale rules.

(page 3, Table 1)

  1. Ensure that Figure 1 is renamed with a descriptive title other than just "PRISMA."

A: Thanks for your comment. We renamed the Figure 1 accordingly.

Figure 1. Preferred Reporting Items for Systematic Review and Meta-Analysis (PRISMA) flowchart for the searching and identification of included studies

  1. Upon reviewing the references, it appears that you have cited MORE THAN TEN references from your own previous work. It is essential to note that self-citation without a valid scientific reason is considered unethical. Therefore, I recommend carefully examining the references and eliminating any instances of unnecessary self-citation to ensure the integrity of the scholarly work.

A: thanks for your comment. We confirm that all cited references contribute to the scholarly content of the paper and they are needed to understand the background of the work. In detail, some references are necessary to understand the topic and the results already reported in the literature. Other references better explain the methodology adopted in the current study in terms of the statistical analysis techniques, data aggregation, and their correct interpretation. However, we agree with your comment and warning, and we revised the references deleting 3 of our previous work.

Reviewer 3 Report

Comments and Suggestions for Authors Periprosthetic infection is a severe complication of joint replacement with low, but stable incidence, even in the most competent clinics. Fortunately, most of the cases of severe periprosthetic infection are sporadic and accumulation of the knowledge and experience in prevention, diagnostics and treatment of this complication is essential for the better practice in this specific issue. According to these statements, the main question addressed by the research was regarded to the outcomes of knee arthrodesis after periprosthetic infection and to whether  the differences in clinical outcomes and complication rates among patients treated by  intramedullary nailing, external fixation and compression plating are comparable. According to the information presented in the introduction part of the paper and as far as reviewer knows, the study gives an original view on the problem and relevant for the field of orthopedic surgery, specifically - to joint arthroplasty, and addressed to the lack of comparative data. The authors conducted meticulous analysis of the current literature regarding knee arthrodesis ater periprosthetic infection. As the result of the systematic search they found 23 papers compatible with the inclusion criteria. The papers were analysed according to the proposed protocol, focused on aetiology, comorbidities, results of treatment, and follow-ups. As the result of the study the authors summarised the data with the conclusions regarding the important points of clinical course and consequences of treatment. Some important findings from quite a large cohort of accumulated cases published in those 23 studies were done. The paper is well designed according to the basic principles of systematic review, the analysed data are confirmed as related to the high-quality studies. The most important information extracted from the original studies is presented in several clearly organised tables. The summarising information is also present in well structured tables, which makes the paper easy and clear for reading. The conclusions are clearly narrated and definitely based on the analysed material. All main questions posed were addressed and systematic search and analysis. The references used and listed are appropriate to the research question and protocol. The paper gives a good impression and should be helpful for the clinical practitioners and scientists. The paper is recommended for the publication in the present version.

Author Response

Reviewer #3

Periprosthetic infection is a severe complication of joint replacement with low, but stable incidence, even in the most competent clinics. Fortunately, most of the cases of severe periprosthetic infection are sporadic and accumulation of the knowledge and experience in prevention, diagnostics and treatment of this complication is essential for the better practice in this specific issue. According to these statements, the main question addressed by the research was regarded to the outcomes of knee arthrodesis after periprosthetic infection and to whether the differences in clinical outcomes and complication rates among patients treated by  intramedullary nailing, external fixation and compression plating are comparable. According to the information presented in the introduction part of the paper and as far as reviewer knows, the study gives an original view on the problem and relevant for the field of orthopedic surgery, specifically - to joint arthroplasty, and addressed to the lack of comparative data. The authors conducted meticulous analysis of the current literature regarding knee arthrodesis ater periprosthetic infection. As the result of the systematic search they found 23 papers compatible with the inclusion criteria. The papers were analysed according to the proposed protocol, focused on aetiology, comorbidities, results of treatment, and follow-ups. As the result of the study the authors summarised the data with the conclusions regarding the important points of clinical course and consequences of treatment. Some important findings from quite a large cohort of accumulated cases published in those 23 studies were done. The paper is well designed according to the basic principles of systematic review, the analysed data are confirmed as related to the high-quality studies. The most important information extracted from the original studies is presented in several clearly organised tables. The summarising information is also present in well structured tables, which makes the paper easy and clear for reading. The conclusions are clearly narrated and definitely based on the analysed material. All main questions posed were addressed and systematic search and analysis. The references used and listed are appropriate to the research question and protocol. The paper gives a good impression and should be helpful for the clinical practitioners and scientists. The paper is recommended for the publication in the present version.

A: Thanks for your comment. We are pleased that our work has been appreciated and that we have proposed a topic of interest to readers of the Journal.

Reviewer 4 Report

Comments and Suggestions for Authors

This systematic review aimed to investigate the outcomes of KA after PJI of the knee. It compared differences in clinical outcomes and complication rates among patients treated by IMN, EF, and CP.

L62: Why "published in December 2023"? Do you mean until December 2023? Also, now it is March? Are there no more articles published until today?

Convert pages 5-8 to landscape.

Discussion: Start the discussion with the aim(s) of the systematic review first and then introduce the findings.

Comments on the Quality of English Language

Good level of English language. Minor corrections are needed.

Author Response

Reviewer #4

This systematic review aimed to investigate the outcomes of KA after PJI of the knee. It compared differences in clinical outcomes and complication rates among patients treated by IMN, EF, and CP.

A: Thanks for your comment. We revised the text point-to-point according to your valuable suggestions.

L62: Why "published in December 2023"? Do you mean until December 2023? Also, now it is March? Are there no more articles published until today?

A: Thanks for your comment. We performed a new systematic revision in the databases we used previously, and no other studies should be included. Therefore, we updated the data to March 2024 and revised the text accordingly.

The PubMed, MEDLINE, Scopus, and Cochrane Central databases were searched in March 2024.

(page 2, lines 67-68)

Convert pages 5-8 to landscape.

A: Thanks for your comment. We provided a revised Table 2.

Discussion: Start the discussion with the aim(s) of the systematic review first and then introduce the findings.

A: Thanks for your comment. We agree with your comment and revised the text accordingly.

The aim of this systematic review was to investigate the outcomes of KA after PJI of the knee. We noted that, the most common causative pathogens causing PJI treated with KA were Staphylococcus species, and this surgical procedure confirmed the effectiveness of treating the pain associated with PJI.

(page 11, line 276-279).